# Accumulation of Agmatine, Spermidine, and Spermine in Sprouts and Microgreens of Alfalfa, Fenugreek, Lentil, and Daikon Radish

**DOI:** 10.3390/foods9050547

**Published:** 2020-05-01

**Authors:** Irena Kralj Cigić, Sašo Rupnik, Tjaša Rijavec, Nataša Poklar Ulrih, Blaž Cigić

**Affiliations:** 1Faculty of Chemistry and Chemical Technology, University of Ljubljana, Večna pot 113, SI-1000 Ljubljana, Slovenia; irena.kralj-cigic@fkkt.uni-lj.si (I.K.C.); tjasa.rijavec@fkkt.uni-lj.si (T.R.); 2Biotechnical Faculty, University of Ljubljana, Jamnikarjeva 101, SI-1000 Ljubljana, Slovenia; saso.vet@gmail.com (S.R.); natasa.poklar@bf.uni-lj.si (N.P.U.)

**Keywords:** polyamines, biogenic amines, germination, medicago sativa, trigonella foenum-graecum, lens culinaris, raphanus sativus, diamine oxidase

## Abstract

Sprouts and microgreens are a rich source of various bioactive compounds. Seeds of lentil, fenugreek, alfalfa, and daikon radish seeds were germinated and the contents of the polyamines agmatine (AGM), putrescine (PUT), cadaverine (CAD), spermidine (SPD), and spermine (SPM) in ungerminated seeds, sprouts, and microgreens were determined. In general, sprouting led to the accumulation of the total polyamine content. The highest levels of AGM (5392 mg/kg) were found in alfalfa microgreens, PUT (1079 mg/kg) and CAD (3563 mg/kg) in fenugreek sprouts, SPD (579 mg/kg) in lentil microgreens, and SPM (922 mg/kg) in fenugreek microgreens. A large increase in CAD content was observed in all three legume sprouts. Conversely, the nutritionally beneficial polyamines AGM, SPD, and SPM were accumulated in microgreens, while their contents of CAD were significantly lower. In contrast, daikon radish sprouts exhibited a nutritionally better profile of polyamines than the microgreens. Freezing and thawing of legume sprouts resulted in significant degradation of CAD, PUT, and AGM by endogenous diamine oxidases. The enzymatic potential of fenugreek sprouts can be used to degrade exogenous PUT, CAD, and tyramine at pH values above 5.

## 1. Introduction

Sprouts and microgreens are popular and trendy foods [1]. The wide variety, available even in a relatively small surface area, offers the opportunity to practice urban gardening under space-limited conditions in houses and apartments. The popularity of sprouts and microgreens is related to their high aesthetic potential and intense taste. The sensory and nutritional properties depend on the content of secondary metabolites [2]. The results of various studies have shown that the content of many bioactive compounds increases significantly during sprouting and in the microgreens [3]. High contents of polyphenols, anthocyanins, and other redox-active compounds, i.e., vitamins, glucosinolates, and minerals, have been found [4,5,6,7]. Modulation of the light regime [8] and the composition of the growth solution for microgreens have shown a high potential for biofortification with minerals [9] and secondary metabolites [10].

Polyamines, such as agmatine (AGM), putrescine (PUT), cadaverine (CAD), spermine (SPM), and spermidine (SPD) are secondary metabolites with two or more amino groups that are closely related to plant growth and development, stabilization of cellular structures, and stress resistance [11]. Increased endogenous synthesis as well as the exogenous application of polyamines, improve seed germination and growth [12]. The results of several studies show that germination leads to a change in the content and profile of polyamines. In the soybean, germination leads to an accumulation of all analyzed polyamines. Maximum values were determined after 48 h, followed by slightly lower values after 96 h of germination. These are still three-fold higher than in ungerminated seeds [13]. The content of all analyzed polyamines increased during germination in lupin sprouts whereas, in fenugreek, only PUT and CAD accumulated, while SPM and SPD remained constant [14]. There are some other reports of changes in polyamine content during germination of legume seeds, where a larger increase of PUT and CAD, compared to SPM and SPD, was observed when dry weight is assumed [15,16]. Accumulation of all polyamines was observed in germinated corn [17] and a large increase in agmatine content in radish [16] and flaxseed sprouts [18]. Reports related to the polyamine transformation in microgreens are rare. In lettuce, a gradual decrease in free SPM and SPD was observed from the microgreens stage (2 weeks) to commercial maturity (10 weeks) [19].

The polyamines, which accumulate in the germinating seeds, not only have intracellular functions but can also serve as a substrate for diamine oxidases. The enzymatic oxidation of predominantly PUT and CAD produces H_2_O_2_, which is involved in cell wall differentiation and programmed cell death and has direct antimicrobial activity when tissue integrity is broken [20,21]. Copper amine oxidases (CuAO) are diamine oxidases with copper ion in the active site and are expressed at high levels in legumes [22]. They are localized either in the apoplasts, in the intercellular spaces, or loosely bound to the cell walls [23,24]. Diamine oxidases are expressed in various tissues of germinated seeds of the *Leguminosae* family. In soybean sprouts, the enzyme is predominantly expressed in the hypocotyl and root system. The activity in bean sprouts has been found mainly in the cotyledons [25], and, in fava beans, in all parts except the cotyledons [26]. The higher enzyme activities were found to be correlated with higher contents of CAD or PUT [27] in the hypocotyl and root of chickpeas. Enzymes that catalyze the oxidative deamination of biogenic amines can be used as dietary supplements. Diamine oxidases of animal origin, incorporated in capsules, can be consumed in the intestinal tract for more efficient oxidation of undesirable dietary biogenic amines. Such treatment effectively reduces the severity of migraine episodes [28]. On the other hand, excessive oxidation of polyamines in the digestive tract is problematic, as the H_2_O_2_ generated is toxic to the intestinal cells. A dietary supplement with a combination of white pea diamine oxidase with catalase, which catalyzes the decomposition of H_2_O_2_ generated by diamine oxidase, resulting in reduced toxicity [29]. The direct oxidation of biogenic amines in the food matrix, prior to ingestion, could be a viable alternative to the use of amine oxidases as dietary supplements.

From the published results, it can be concluded that the content of polyamines generally increases during sprouting. From a nutritional point of view, there is no simple answer to whether this is beneficial or not. Large contents of PUT and CAD that accumulate as a result of endogenous synthesis in plants or by microbial decarboxylation of amino acids [30] are certainly not desirable. These foul-smelling compounds are slightly toxic to the intestinal cells [31] and, mainly interfere with the enzymatic oxidation of tyramine (TYR) and histamine (HIS) in the digestive tract [32], which increases their negative effects. Dietary intake of AGM, SPM, and SPD may be desirable. SPM and SPD, in particular, appear to have cardioprotective and neuroprotective effects [33]. AGM, which can cross the blood–brain barrier, can be consumed in large quantities without adverse health effects and can relieve the symptoms of central nervous system disorders, including major depression [34]. Endogenous synthesis of polyamines in mammals decreases with age [35], and dietary intake of polyamines, particularly SPD, is directly related to lower mortality, as has been found in a prospective population-based study [36]. However, dietary polyamines are a double-edged sword, as they can potentiate the growth of certain cancers, most probably due to the stabilizing of DNA [37]. Dietary intake of polyamines is therefore generally desirable, but should also be controlled because of the possible adverse effects. The content of SPM, SPD [38], and AGM [39] in the diet is becoming an important issue. Different seeds and meats are quantitatively the main sources of SPM and SPD, while certain types of fermented foods are rich in AGM (Table 1).

The objectives of the present study are (I) to determine the polyamine content in seeds, sprouts and microgreens of three legumes and one cruciferous plant, (II) to evaluate whether microgreens are nutritionally superior to sprouts in terms of polyamine content, and (III) to evaluate the enzymatic potential of sprouts to degrade undesirable biogenic amines.

## 2. Materials and Methods

### 2.1. Materials

Acetonitrile (gradient HPLC grade) was obtained from Fischer Scientific (Hampton, NH, USA). Ultrapure water was obtained with a Milli-Q water system (Millipore Merck, Darmstadt, Germany). Acetone (≥99.8%), n-hexane (≥95%), HCl (37%) were obtained from Honeywell (Charlotte, NC, USA), NaOH (p.a.), NH_3_ (25%), acetic acid (glacial), NaH_2_PO_4_ × 2H_2_O (p.a.) and NaHCO_3_ (p.a.) from Merck (Darmstadt, Germany). Dansyl chloride (≥99%) and amines were obtained from Sigma-Aldrich (St. Louis, MO, USA): 1,7-diaminoheptane (98%), agmatine sulfate (≥97%), phenethylamine (99%), histamine (≥97%), cadaverine (≥96.5%), putrescine (≥98.5%), spermidine (≥98%), spermine (≥97%), tyramine (≥98.5%) and tryptamine (TRP) (≥98%).

Fenugreek (*Trigonella foenum-graecum*), lentil (*Lens esculentum*), alfalfa (*Medicago sativa*), and daikon radish (*Raphanus sativus*) seeds designated for sprouting were supplied by Amarant (Kresnice, Slovenia).

### 2.2. Seed Sprouting

The seeds of each species were rinsed, then soaked in tap water (23 °C, pH 7.5, 450 µS/cm) at room temperature for 6 h. The soaked seeds were germinated in Schnitzer (Offenburg, Germany) sprouting trays (diameter 18.5 cm, depth 4 cm) with a built-in drainage system. Four trays, each containing different seeds, were organized in a vertical tower to retain moisture. Every 8 h, the tower was dismantled, and the germinated seeds were rinsed in separate trays with tap water (approx. 500 cm^3^ per tray) to prevent microbial spoilage. After the water had been drained off, the tower was reassembled. The germinated seeds were incubated at 23 ± 1.5 °C for 4 days. The seeds of all four species were germinated three times (independent experiments).

### 2.3. Growing Microgreens

The seeds were soaked, as described in Section 2.2, and spread on a fully hydrated Urbanscape rockwool slab 12 × 12 × 2 cm (Knauf insulation, Škofja Loka, Slovenia) with half-strength Hoagland nutrient solution [57]. The hydrated slabs were placed separately in plastic trays and filed up to a height of 1 cm with half-strength Hoagland nutrient solution. The germinated seeds were incubated for 10 days at 23 ± 1.5 °C (relative humidity 60%) under 16 h/8 h (light/dark cycle) photoperiod and a photon flux density of 36 µmol m^−2^s^−1^ provided by cool-white fluorescent lamps MASTER TL D 58W/840 (Philips, Amsterdam, The Netherlands). The germinating seeds and later the seedlings were moistened once a day by spraying with distilled water. The loss of solution in the trays was compensated by a daily addition of distilled water. Microgreens of all four species were grown four times (independent experiments).

### 2.4. Sample Preparation

#### 2.4.1. Extraction Procedure

All sprouts and microgreens were homogenized fresh, unless otherwise indicated. Approximately 1.5 g (known mass) of sprouts or microgreens (upper two-thirds of the seedling height) were weighed into 50 mL polypropylene centrifuge tubes, filled with 15 mL of 0.4 M HCl containing 10 mg/mL of 1,7-diaminoheptane (IS) and immediately homogenized (30 s homogenization/30 s resting time-repeated 3 times) with the T-25 Ultra-Turrax (Ika-Labortechnik, Staufen, Germany) at 13,500 rpm. The ungerminated seeds were homogenized in the same way as sprouts and microgreens. The homogenized samples were left at room temperature for 5 min and then centrifuged at 4000× *g* for 5 min. Aliquots of the partially cleared homogenates were transferred to 2 mL centrifuge tubes and further centrifuged at 15,000× *g* for another 5 min. The supernatant was transferred to new centrifuge tubes and used for derivatisation, which was performed within 2 h after homogenization. The polyamines extracted in 0.4 M HCl can also be stored at −20 °C for one week, as this storage had no influence on the determined polyamine content.

The moisture content of sprouts and microgreens was determined by oven drying the samples at 105 °C to constant weight (≈6 h).

#### 2.4.2. Freezing and Thawing

Liquid nitrogen was poured over the sprouts to induce immediate freezing. Frozen sprouts were immediately transferred into polypropylene bags and stored at −20 °C. One week storage at −20 °C of frozen sprouts did not result in lower polyamine content if they were immediately transferred to 0.4 M HCl containing 10 mg/L IS and homogenized as explained in Section 2.4.1. To assess the influence of thawing on the polyamine content, frozen sprouts were evenly spread on a glass Petri dish and homogenized in 0.4 M HCl containing 10 mg/L of IS after 5, 20, 60, and 180 min of thawing at room temperature.

#### 2.4.3. Fenugreek Sprouts as a Source of Amine Oxidases

Fresh fenugreek sprouts (5 g) were homogenized (30 s homogenization/30 s resting time-repeated twice) with the T-25 Ultra-Turrax (Ika-Labortechnik, Staufen, Germany) at 13,500 rpm in 25 mL of MQ water. 4 mL of fresh homogenates were transferred into 80 mL glass beakers containing a mixture of 100 mM buffer with suitable pH (6 mL) and 10 mL of a mixture of biogenic amines. Buffers (100 mM) with pH 4 and 5 were prepared previously from acetic acid with the addition of NaOH. Buffers (100 mM) with pH 6, 7, and 8 were prepared from sodium dihydrogen phosphate with the addition of NaOH. The mixture of polyamines (100 mg/L of CAD, HIS, PHE, PUT, TRP, and TYR) was previously adjusted to pH 7 by the addition of HCl solution. The concentration of individual polyamines in the reaction mixtures was 50 mg/L, 30 mM for buffer, and 33 g/L for fenugreek sprouts (assuming that the densities of all solutions/sprouts are approximately 1 g/mL). The reaction mixtures were incubated at 25 °C on a magnetic stirrer at a stirring speed of 250 min^−1^.

At specified time intervals (2, 5, 12, 25, 60, and 120 min), 750 µL of the reaction mixtures were transferred into 2 mL centrifuge tubes containing 750 µL of IS (20 mg/L) in 0.8 M HCl, thoroughly mixed to stop the reaction, centrifuged and proceed as described in Section 2.4.1.

As the pKa values of all amino groups (except the imidazole group of HIS) are above 9, the neutral solution of polyamines had no significant influence on the pH value of the reaction mixtures. The pH value of the reaction mixtures was also checked at the end of the incubation period (120 min), and we found that it did not differ from the initial pH value by more than for ±0.1. Buffers at the concentration used in the reaction mixture did not seem to affect the derivatization yield of the biogenic amines.

### 2.5. Preparation of Standard Solutions and Derivatization

#### 2.5.1. Internal Standard

1,7-diaminoheptane (IS) was used as an internal standard in amine standard solutions and at various levels of sample preparation to control all steps of sample manipulation from homogenization, derivatization, and injection into HPLC. Stock IS solution with a concentration of 1.0 g/L was prepared by weighing 10 mg of IS and dissolving it in 10 mL of 0.4 M HCl or 0.8 M HCl.

#### 2.5.2. Amine Standards and Calibration Solutions

Standard solutions of individual amine (AGM sulfate, TRP, PEA, PUT, CAD, HIS, TYR, SPD, and SPM) were prepared with a concentration of 1.0 g/L. Then, 10 mg of solid amines (AGM sulphate, TRP, HIS, TYR, and SPM) were dissolved in 10 mL of 0.4 M HCl solution containing 10 mg/L of IS. Afterwards, 10 μL of liquid amines (CAD, PUT, SPD, PEA) were pipetted and dissolved in different amounts of 0.4 M HCl with IS, according to their density (*ρ*(CAD) 0.873 g/mL, *ρ*(PUT) 0.877 g/mL, *ρ*(SPD) 0.925 g/mL, *ρ*(PEA) 0.962 g/mL). Mixed calibration standard solutions containing all 9 amine compounds were prepared in the concentration range of 0.3–45.0 mg/L, with a 0.4 M HCl solution containing 10 mg/L of IS.

#### 2.5.3. Derivatization Procedure with Dansyl Chloride (DNS–Cl)

The solution of DNS–Cl with a concentration of 10 g/L was prepared in acetone. The derivatization was performed in a 1.5 mL centrifuge tube, as previously described [58]. Then, 250 μL of the calibration solution or sample was pipetted, and then 50 μL of 2 M NaOH, 75 μL of the saturated solution of NaHCO_3_, and 500 μL of DNS–Cl solution were added, each addition followed by vortexing. The derivatization was carried out in a heating block at 40 °C for 60 min. After incubation, 25 μL of a 25% aqueous NH_3_ solution was added to the solution and left at room temperature for 30 min. Afterwards, 350 μL of acetone was added, the solution was mixed again, and filtered through a 0.45-μm nylon filter before HPLC analysis.

### 2.6. HPLC Analyses

The HPLC determinations were performed with UV–vis and fluorescence detectors. All chromatograms were recorded using both detectors (Figure 1). Due to better sensitivity and selectivity of the dansylated amines obtained by a fluorescence detector, the signals for the latter were used for peak-area integration and further evaluation. The only exception was histamine, where spectrophotometric signals were employed because the fluorescence yield of its dansylated derivative was low. Across all samples, the ratio of peak areas in the chromatograms was constant by the respective detectors for the individual amine derivative, which corroborated the supposition that the integrated peak actually reflected the content of the amine analyzed. All peak areas were normalized to those of IS. The matrix showed only a small influence on the derivatization yield of IS. The median derivatization yield of IS in the complex matrix was 86% (upper quartile 91% and lower quartile 80%).

Instrumentation: Agilent HPLC system 1100 (Palo Alto, CA, USA), equipped with a degasser, a quaternary pump, an autosampler, a UV–vis and a fluorescent detector was used. The wavelength of the UV–vis detector was 254 nm, the excitation wavelength of the fluorescence detector was 350 nm, and the emission wavelength 520 nm. A Kinetex XB-C18 (5 μm, 100 Å, 150 × 4.6 mm) column with a guard column of the same particle size was used (Phenomenex, Torrence, CA, USA). The flow rate of the mobile phase was 0.7 mL/min. The separation was performed with a gradient of two eluents. Eluent A was MQ water and eluent B was acetonitrile. The initial composition of the mobile phase was 40% B, which changed linearly from 0 to 25 min to 80% B. At 25 to 30 min, a second linear gradient was used to change the mobile phase from 80% B to 100%, where it remained constant until 35 min. Then the composition changed linearly within 5 min to the initial 40% B. The column was then equilibrated for 2 min.

### 2.7. Statistical Analysis

A non-parametric Mann–Whitney test [59,60] based on the data ranking was used for the statistical analysis. The differences in the content of a particular polyamine in ungerminated seeds, sprouts and microgreens were significant at the *p* < 0.05 level.

## 3. Results and Discussion

### 3.1. Polyamine Content in Sprouts and Microgreens of Lentil, Fenugreek, Alfalfa, and Daikon Radish

Polyamine contents were determined in different stages of plant growth for four different species—lentil, fenugreek, alfalfa, and daikon radish. Sprouting and microgreen formation resulted in a large transformation of polyamines in all species analyzed. In general, a large accumulation of polyamines was observed. In each species, the content of at least one of the polyamines increased by two orders of magnitude compared to that in ungerminated seeds. The changes in polyamine content that occurred during growth from seed to sprout, and, finally, to microgreen, were specific to each species, so the results are presented separately. A comparison of the polyamine content in the different species and growth stages was carried out in terms of dry weight (DW) values.

#### 3.1.1. Lentil

The contents of AGM (38 mg/kg), PUT (50 mg/kg), CAD (2 mg/kg), SPD (101 mg/kg), and SPM (35 mg/kg) and their relative proportions in lentil seeds were in similar ranges to those previously observed [16]. Sprouting only led to an increase in the contents of PUT (238 mg/kg) and CAD (742 mg/kg), as in previous germination experiments with lentil [16]. A different scenario was observed in soybeans [13], in which the content of all polyamines in sprouts increased. Since both PUT and CAD are nutritionally unfavorable, the nutritional value of the sprout in terms of polyamine composition was lower than that of ungerminated seeds. As the polyamine content was reported on a DW basis, it was still several times lower than that found in certain fermented foods of plant and animal origin [41] and was therefore highly unlikely to pose a health risk. The separate analysis of the polyamines in the epicotyl (part of the seedling above the cotyledons) and in the hypocotyl (part of the seedling below the cotyledons) of sprouts showed that the polyamine content in the epicotyl was an order of magnitude lower. In addition, a large difference in the spatial distribution of CAD in chickpea seedlings was found previously [27].

It was found that the content of CAD (47 mg/kg) in microgreens was one order of magnitude lower than in sprouts (Figure 2a), which is consistent with the spatial distribution in sprouts. CAD was the only polyamine with a lower content in lentil microgreens than in sprouts, while values four times higher were found for SPD (579 mg/kg) and three times higher for SPM (88 mg/kg). Levels above 500 mg/kg SPD are extremely high compared to that in other foods. A similar concentration range was only found for some mature cheddar cheeses [61], mushrooms [62], and germinated flaxseeds [18]. The estimated average daily dietary intake of SPD for the USA population [63] is achieved by the consumption of 100 g lentil microgreens (on a fresh weight basis (FW)). The higher contents of SPD and SPM and lower content of CAD indicate that lentil microgreens are nutritionally superior to sprouts in terms of polyamine composition.

#### 3.1.2. Fenugreek

Fenugreek is a traditional medicinal plant found in many cultures around the world [64]. Its culinary uses include the use of seeds as a spice and fresh leaves or sprouts and microgreens, which have a mild but slightly bitter taste. The absolute contents of AGM (2 mg/kg), PUT (2 mg/kg), SPD (59 mg/kg), and SPM (11 mg/kg) were lower in ungreminated seeds (Figure 2b, Figure A3) than in lentil seeds (Figure 2a). They were in a similar range as in the literature [14], except for SPD, in which a higher amount was determined in current work. Sprouting led to a large increase in practically all polyamines. CAD was below the limit of detection in ungerminated seeds and accumulated to more than 3000 mg/kg DW during sprouting. Such high levels are unusual in food samples and are twice as high as the maximal levels found in some acid-cured cheeses [65] or fresh scallops [62]. Higher DW based values were only found in some fermented soybean sauces [66]. AGM (1006 mg/kg) and PUT (1079 mg/kg) levels were above 1000 mg/kg DW but PUT levels were still in the range of foods such as sauerkraut [67], aged cheeses and even freshly squeezed citrus juices, which are among the richest sources of polyamines in unfermented foods [40]. Little information has been published on the AGM content of foods, but levels above 1000 mg/kg DW are undoubtedly high, as higher levels are found only in some fermented soybean products [39]. Despite initially lower contents in seeds, SPM (240 mg/kg) and SPD (87 mg/kg) in fenugreek sprouts have accumulated to twice the levels found in lentil sprouts. The accumulation of SPM and SPD during germination has not been observed before [68], and the increases of PUT and CAD contents were much less pronounced. The reason for the observed differences could be either their biological origin or the growth regime or methodology. In the previous study, the samples were processed by freeze-drying, which may have influenced the profile of the polyamine content, as shown in Section 3.2. Due to the extremely high content of CAD (3563 mg/kg), fenugreek sprouts cannot be considered as a healthy food, despite the high amounts of the nutritionally beneficial polyamines AGM, SPD, and SPM.

The composition of the polyamines in fenugreek microgreens has been improved in terms of nutritional value, as well as in lentils. The four-fold lower content of CAD (873 mg/kg) and an order of magnitude higher content of SPM (922 mg/kg) in microgreens were the most pronounced changes compared to sprouts. None of the foods listed in the published databases [38,40,62] showed similarly high levels of SPM on a DW basis as we found in fenugreek microgreens. The estimated average daily dietary intake of SPM for the USA population [63] is exceeded by the consumption of 90 g of fenugreek microgreens (based on FW).

#### 3.1.3. Alfalfa

Alfalfa, fenugreek, and lentil all belong to the legume family. The contents of AGM (11 mg/kg), PUT (12 mg/kg), CAD (1 mg/kg), SPD (140 mg/kg), and SPM (34 mg/kg) in ungerminated alfalfa seeds were lower than those for lentil, but higher than for fenugreek. Similar to lentil sprouting, alfalfa sprouting did not result in a higher content of SPM (34 mg/kg), while the content of SPD (323 mg/kg) doubled compared to ungerminated seeds. Sprouting resulted in two orders of magnitude greater contents (Figure 2c) of AGM (2669 mg/kg) and PUT (1015 mg/kg), and the accumulation of CAD (1910 mg/kg) resulted in higher levels than that for lentil sprouts and lower levels than for fenugreek. Polyamines PUT and CAD present in sprout samples could be formed as a result of microbial spoilage [16] or endogenous synthesis by plants. Insufficient hygienic standards could impair microbial spoilage. However, even with 11 log colony-forming units/g and five orders of magnitude higher microbial load after the end of the 12-day storage period of four sprout varieties, only a minor increase in CAD and PUT content was observed [69]. In order to test whether some polyamines were outside the plant cells, due to microbial spoilage, the sprouts were rinsed with extraction buffer (30 s), and only about 10% polyamines were found in such extracts. The ratios of the polyamine contents reflected those in sprouts, strongly suggesting that polyamines in the “rinsing solution” originated from ruptured cells (0.4 M HCl).

The contents of PUT (1034 mg/kg), SPD (292 mg/kg), and SPM (30 mg/kg) in microgreens remained in similar ranges to those in sprouts. The content of CAD (257 mg/kg) was lower by an order of magnitude, as was also observed in lentil and fenugreek microgreens. Alfalfa sprouts, and especially, microgreens (Figure 2c), stood out as extremely rich sources of AGM (5392 mg/kg). Such high contents were found only in some samples of fermented soybean pastes [70]. As noted above, there are relatively few data on the content of AGM in foods [39]. This is rather surprising, considering the many health benefits of AGM supplements and the fact that long-term intake of relatively high doses of AGM should be safe [53]. Dietary AGM can be absorbed in the small intestine and pass the blood–brain barrier [71], where it can interfere with some important central nervous system disorders. The majority of the experiments were conducted in a rodent model with ingested daily doses of 10 mg/kg or higher [34]. Human trials are rare; daily doses of about 10 mg/kg [44] have been shown to be effective in relieving pain in lumbar disc-associated radiculopathy. An intake of more than 1 kg of alfalfa microgreens (FW) would be necessary to reach an equivalent level, which is unrealistic. However, some rodent experiments [42] have shown that oral administration of 0.1 mg/kg of AGM alone produces the antidepressant effects. If similar results are confirmed in human studies, this would put a different perspective on alfalfa sprouts as a source of dietary AGM.

#### 3.1.4. Daikon Radish

The absolute content of AGM (52 mg/kg), PUT (11 mg/kg), CAD (4 mg/kg), SPD (96 mg/kg), and SPM (56 mg/kg) in ungerminated seeds of daikon radish was similar to that in lentil seeds. The germination of this cruciferous vegetable (Figure 2d) resulted in some differences from the analyzed legumes. There was only a slight increase in the content of CAD (21 mg/kg), and the values in sprouts were one or two orders of magnitude lower than in legume sprouts. On the other hand, the contents of AGM (4270 mg/kg) and SPD (423 mg/kg) in daikon sprouts were the highest of all four plants analyzed. In combination with the relatively low content of CAD, this indicates that daikon radish sprouts are an excellent source of nutritionally beneficial polyamines. Previous reports on the change in polyamine content during sprouting [16,72], were consistent with the current results and showed a large accumulation of AGM. In the same studies, it was found that SPD content in sprouts decreased on an FW basis, but, assuming 10% DW, the recalculated values were consistent with the observations of our study. An increase in SPD content was also observed in [73], but the reported levels are an order of magnitude lower in both ungerminated seeds and sprouts.

In contrast to legumes, in which microgreens were nutritionally better than sprouts, due to a lower content of CAD and a higher content of AGM, SPD or SPM, daikon radish microgreens were nutritionally inferior to sprouts. Contents of the nutritionally beneficial polyamines AGM (3951 mg/kg), SPD (250 mg/kg), and SPM (17 mg/kg) were lower in microgreens than in sprouts. The content of SPM in microgreens was even lower than in ungerminated seeds.

### 3.2. The Effect of Thawing on The Determined Polyamine Contents in Frozen Sprouts

The general protocol for the extraction of biogenic amines, described in materials and methods (Section 2.4.1), explicitly requires immediate extraction of freshly harvested microgreens and sprouts. The reason for this rigor is that the freeze/thaw cycle initiates extensive transformations of polyamines. Consequently, the polyamine contents determined depend strongly on the time delay of sampling after the start of thawing (Figure 3, Figure A1). Freezing (in liquid nitrogen) and storage for a few hours at −20 °C was in itself not problematic, but thawing at room temperature was. The contents of the majority of the polyamines analyzed already decreased on a time scale of minutes. In the case of fenugreek (Figure 3), approximately 30% lower contents of PUT and CAD were found in frozen sprouts (−20 °C) that were left at room temperature for only 5 min before homogenization in 0.4 M HCl. During longer thawing times, considerable amounts of SPD and AGM were also degraded. Similar results were observed for lentil and alfalfa sprouts, in which CAD and PUT were most susceptible to degradation (Figure A1a,b). Daikon radish sprouts were much less affected by freeze/thawing (Figure A1c).

Experimental artifacts, resulting from sample preparation, are not unusual and are often related to the enzymatic conversion of metabolites. A typical example is the oxidation of ascorbic acid to dehydroascorbic acid, catalyzed by ascorbate oxidase, which can lead to a distorted ratio of both forms of vitamin C when samples are homogenized under conditions where the enzyme remains active [74]. As shown in Figure 3, the polyamines are also highly susceptible to degradation, which can lead to experimental artifacts if the samples are not handled properly. Indeed, there is little overall consistency in the sample preparation protocols for the analysis of polyamines in sprouts. Fresh sprouts could be ground or homogenized in a grinder before extraction in acid [13,15,16], stored in a frozen state before extraction [75], lyophilized before extraction [68], or directly homogenized in acid [14]. Any protocol that disrupts tissue integrity prior to extraction could result in a significant transformation of polyamines if the criteria for enzyme activity are met.

It has been observed that freezing and thawing of soybean sprouts [76] resulted in significant accumulation of γ-aminobutyric acid (GABA), which is formed in two enzymatic steps from PUT, including the action of amine oxidases as the first step. Recently it was shown that GABA accumulation in thawed sprouts is most probably the result of ruptured cellular structures rather than the higher enzyme activity itself [77]. In neither of these two studies, where such enzymatic activity could be observed with certainty, was the change in polyamine profiles assessed. Since the transformation of polyamines in legume sprouts was much more pronounced than in radish, the most likely candidates are the diamine oxidases with copper ion in the active site, termed copper amine oxidases (CuAOs). CuAOs are expressed in high levels in legumes [22] and are localized in apoplasts, intercellular spaces, or loosely bound to the cell walls [23,24]. The freezing and thawing of plant tissue is detrimental to its structural integrity and leads to rapid changes in the content of some secondary metabolites, often resulting in lower sensory quality and nutritional value [78]. However, this enzymatic potential can also be used to increase the content of desired bioactive constituents [77].

### 3.3. Degradation of Exogeneous Biogenic Amines by Homogenized Sprouts

Copper amine oxidases (CuAO) from the legume family have, in general, optimal activities at neutral or slightly alkaline pH in general and relatively broad substrate specificity, as they catalyze the oxidation of diamines and polyamines, with PUT and CAD as optimal substrates [79,80,81,82]. A similar substrate specificity could be derived from the degradation pattern of polyamines observed in thawed fenugreek (Figure 3), lentil, and alfalfa seeds (Figure A1a,b). According to some older studies [83], the substrate specificity of CuAO could be even broader, including monoamines. The monoamines TYR, PEA, and TRP are often present in fermented foods together with PUT, CAD, and HIS. The content of PEA and TRP in foods is generally low, while HIS and TYR may be present in relatively high concentrations and may have adverse health effects associated with a local immune response (HIS) or increased blood pressure and migraine (TYR). PUT and CAD are less toxic to intestinal cells than TYR and HIS, and only some foods with the highest content [31] could potentially have an adverse effect. Of the sprouts included in the study, even those with the highest CAD/PUT content do not pose a direct risk if consumed in the hydrated form. PUT and CAD are nutritionally problematic, mainly due to their interference with the activity of intestinal amine oxidases with broad substrate specificity, since, at high concentrations of these diamines, less HIS and TYR is oxidized [32]. Even low dietary doses of biogenic amines are problematic for persons taking monoamine oxidase inhibitors as part of their antidepressant therapy [84].

Optionally, biogenic amines can be degraded in the food matrix before consumption. Accordingly, it was investigated as to whether homogenized sprouts could be used as a source of the enzyme for the oxidation of biogenic amines in complex mixtures. Homogenized fenugreek sprouts (less than 0.5% DW in suspension) efficiently oxidized PUT, CAD, and even TYR, when all analyzed biogenic amines were present in the mixture at concentrations of 50 mg/L and in the pH range of 6 to 8. TYR was even more efficiently oxidized than PUT and CAD (Figure 4). After 5 min of incubation at pH 6, 53% of TYR, 32% of PUT, and 22% of CAD were oxidized (Figure 4a–c). At the end of the incubation period (2 h), only traces of these 3 polyamines were observed. At pH 5, the activity was much lower so that after 2 h, the values observed were closely similar to those at pH 6 after 5 min. At pH 4, the enzyme was not active against any of the substrates.

Three other substrates were much less susceptible to oxidation (Figure A2). Under optimal conditions, only 40% of TRP was oxidized after 2 h of incubation, while homogenized sprouts had practically no effect on the stability of HIS and PEA. The substrate specificity of the amine oxidases in fenugreek appears to differ from that of the white pea, for which HIS is a better substrate than TYR [29]. Fenugreek sprouts could, potentially, be used to reduce PUT, CAD, and TYR loads of fermented high protein food products that stabilize the pH of the matrix above 5. Some examples are sausages and cheeses or, when more acidic fermented vegetables and pulses are added to the matrix, with pH above 5. On the other hand, the content of biogenic amines PUT and CAD in legume sprouts could be reduced simply by freezing and thawing for several dozen minutes. Under these conditions, considerable amounts of PUT and CAD are degraded, while the content of nutritionally beneficial SPD is only slightly reduced (Figure 3).

## 4. Conclusions

Germination of lentil, fenugreek, alfalfa, and daikon radish led to the accumulation of their total polyamines. A large increase in CAD content was observed in all three legume sprouts. In the microgreens of these legumes, the CAD content was substantially reduced, whereas here, compared to the sprouts, more AGM, SPD, or SPM accumulate. These are considered to be nutritionally beneficial. In daikon radish sprouts, AGM is the major polyamine, while the formation of microgreens led to a reduction in the content of the nutritionally beneficial polyamines. This behavior was reversed with respect to the changes in the content in legumes.

Tissue damage of legume sprouts led to considerable degradation of polyamines, especially PUT, CAD, and AGM. Experimental artifacts or modulation of the polyamine composition of sprouts are among the possible implications. Homogenized fenugreek sprouts can be used for the degradation of exogenous PUT, CAD, and TYR at pH-values above 5.

## Figures and Tables

**Figure 1 foods-09-00547-f001:**
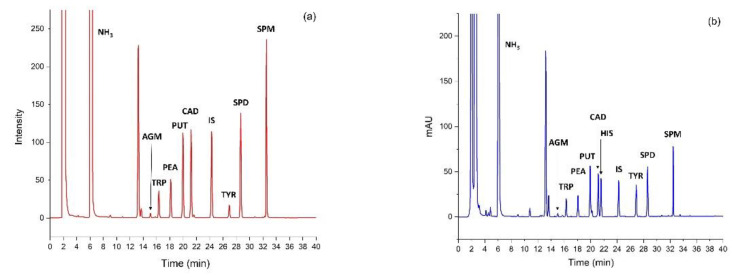
Chromatograms of the standard solution (11 mg/L) of the dansylated biogenic amines agmatine (AGM), tryptamine (TRP), phenethylamine (PHE), putrescine (PUT), cadaverine (CAD), histamine (HIS), 1,7-diaminoheptane (IS), tyramine (TYR), spermidine (SPD), and spermine (SPM). (**a**) fluorescence detector (350/520 nm) and (**b**) UV–vis (254 nm).

**Figure 2 foods-09-00547-f002:**
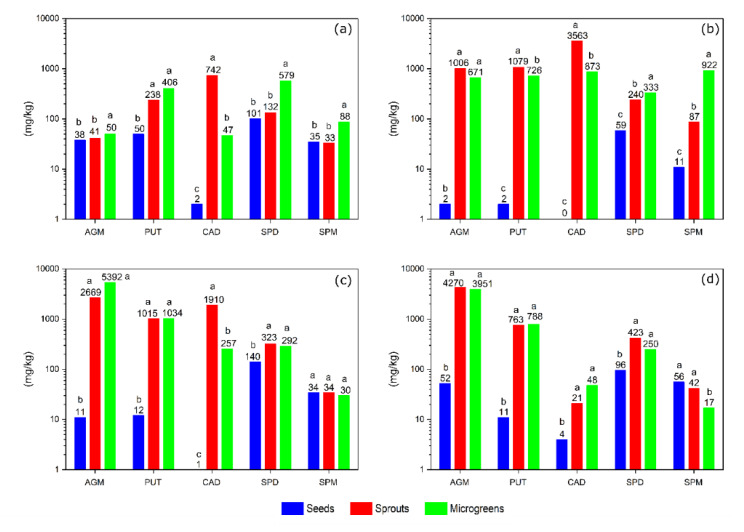
Content of the polyamines agmatine (AGM), putrescine (PUT), cadaverine (CAD), spermidine (SPD), and spermine (SPM) in seeds, sprouts and microgreens of (**a**) lentil, (**b**) fenugreek, (**c**) alfalfa, and (**d**) daikon radish. The data are presented on a logarithmic scale and expressed on a dry weight basis. When the content of a given polyamine in seeds, sprouts, and microgreens differs significantly, it is labeled with different letters.

**Figure 3 foods-09-00547-f003:**
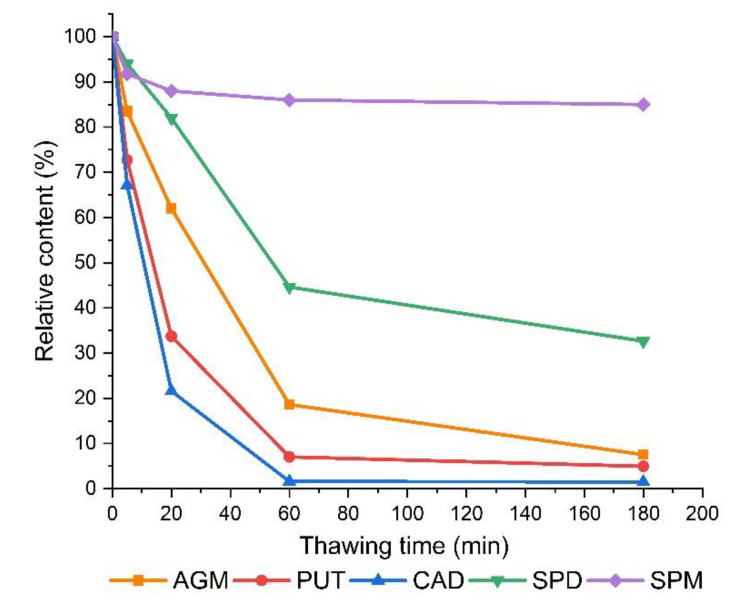
Time-dependent changes in the content of polyamines (agmatine (AGM), putrescine (PUT), cadaverine (CAD), spermidine (SPD), and spermine (SPM)) during thawing of the frozen fenugreek sprouts.

**Figure 4 foods-09-00547-f004:**
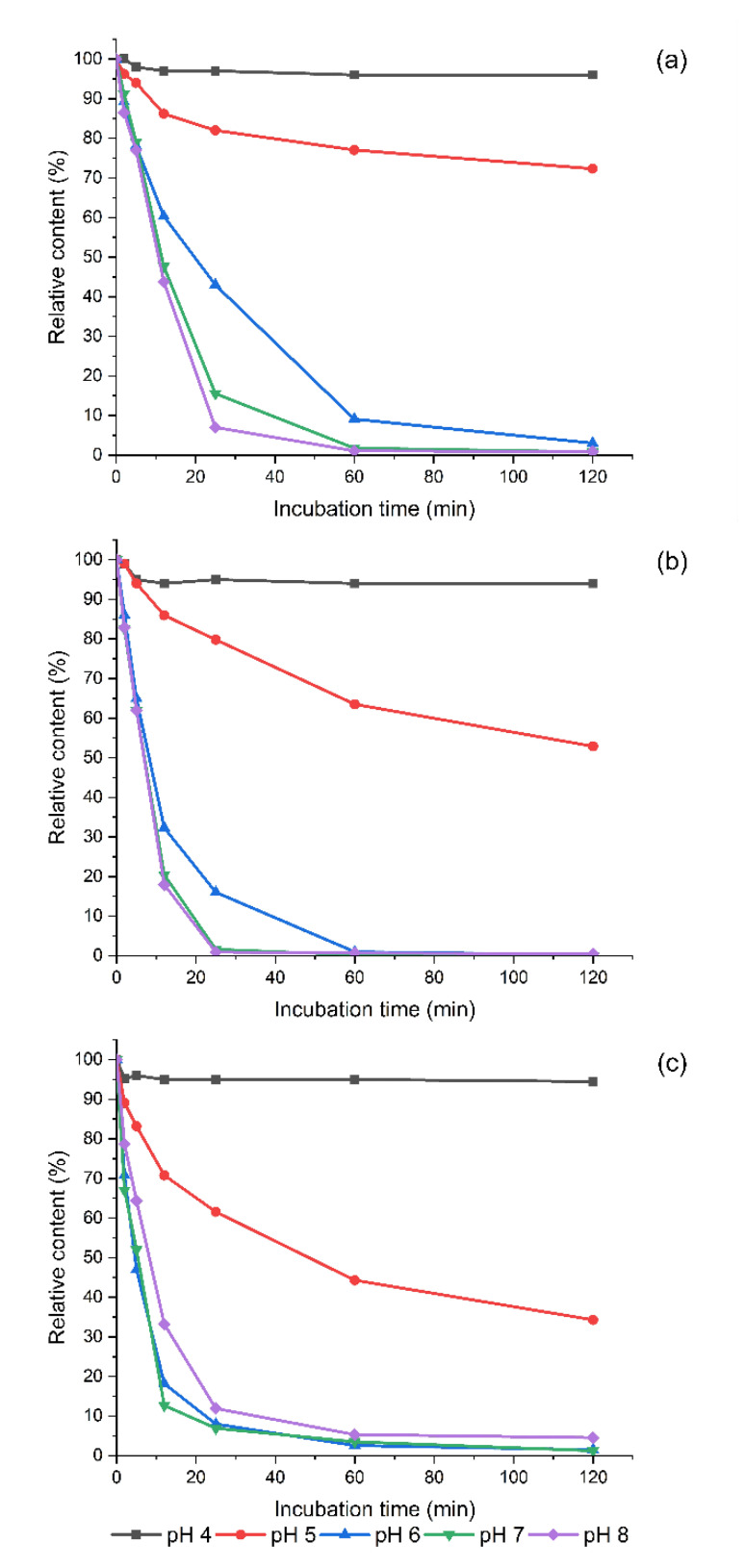
Time-dependent degradation of (**a**) putrescine, (**b**) cadaverine, and (**c**) tyramine by homogenized fenugreek sprouts at different pH values.

**Table 1 foods-09-00547-t001:** Food sources and health effects of dietary polyamines.

	Agmatine (AGM)	Putrescine (PUT)	Cadaverine (CAD)	Spermidine (SPD)	Spermine (SPM)
**Biosynthesis in humans**	Putative From arginine (arginine decarboxylase)	Yes From ornithine/agmatine (ornithine decarboxylase/ agmatinase)	No	Yes From putrescine and S-adenosylmethioninamine (spermidine synthase)	Yes From spermidine and S-adenosylmethioninamine (spermine synthase)
**Main dietary sources**	Fermented food, various sprouts [16,39]	Fermented foods, Citrus fruits and vegetables, legumes [38,40,41]	Fermented foods, legume sprouts [16,42]	Legumes, brassica vegetables, mushrooms, cheese [38,40]	Liver, meat, legumes, brassica vegetables, cheese [38,40]
**Oral intake** **Positive health effects** **Animal/human studies**	Large number of studies showing an antidepressant effect in rodents (0.0001–80 mg/kg) [34,42] Antidepressant effect in humans (2–3 g/day) [43] Pain relief in humans related to radiculopathy (0.75–2 g/day) [44]	Improved intestinal immune function in piglets (10 mg/kg day) [45] Longer survival of mice on high daily polyamine diet (15 mg PUT/kg BW) [46]	/	Longer life span of humans with higher dietary intake of spermidine (5.7 years) [36] Statistically significant increase of life span in mice (0.3 mM in water) [47] Prevention of arterial aging in mice (3 mM in drinking water) [48] Longer survival of mice on high daily polyamine diet (35 mg SPD/kg BW) [46] Daily intake of 29 mg/kg promotes liver regeneration in rats [49]	Statistically significant increase in life span in mice (3 mM in water) [47] Antidiabetic effect on mice at 10 mg/kg daily intake [50] Daily supplement of 80 mg/kg increases the antioxidant capacity rat spleen and liver under oxidative stress [51] Longer survival of mice on high daily polyamine diet (12 mg SPM/kg BW) [46] Daily intake of 10 mg/kg promotes liver regeneration in rats [49] Daily supplement of 80 mg/kg in piglet diet alleviate inflammatory response and enhance the immune function [52]
**Oral intake** **Toxicity and negative health effects** **Animal/human studies**	No adverse effect of 5 years daily agmatine intake of 1.5 g/day for humans [53] and 21 days of 2 g/day for humans [44]	Non-observed adverse effect of daily putrescine or cadaverine intake (180 mg/kg for rats) [54] Formation of nitrosamine [32] Potentiation of histamine and tyramine toxicity [32]		Non-observed adverse effect of daily spermidine intake (60 mg/kg for murines) [55], (83 mg/kg for rats) [54] Promotion of growth of existing tumors in mice [56]	Non-observed adverse effect of daily spermine intake (19 mg/kg for rats) [54]

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
