# Peer review of "Accumulation of Agmatine, Spermidine, and Spermine in Sprouts and Microgreens of Alfalfa, Fenugreek, Lentil, and Daikon Radish"

_foods, 2020, doi:10.3390/foods9050547_

Round 1

Reviewer 1 Report

The authors carried out the current study entitled“Accumulation of agmatine, spermidine and spermine in sprouts and microgreens of alfalfa, fenugreek, lentil and daikon radish “ to 1) determine the polyamine content in seeds, sprouts and microgreens of three legumes and one cruciferous plant, 2) evaluate whether microgreens are nutritionally superior over sprouts in terms of polyamine content and 3) evaluate the enzymatic potential of sprouts to degrade undesired biogenic amines.. The study is on a topic of relevance and general interest to the readers of the journal. I found the paper to be overall well prepared and felt confident that the authors performed careful and thorough experiment and spectral processing. I have several significant concerns about the presentation of the data that should be addressed.

Abstract: Authors need to avoid using personal pronoun (eg. We, our ..etc), please apply this over a;; the manuscript.

Line 16-18: change of PUT to PUT, of SPD to SPD, and of SPM to SPM

Keywords: I recommend the author to use keywords that are differs from the title words

Introduction:

Table 1: some statements need to be supported by reference.

Some sentences are very long page 2 line 63-67 or confusing page 2 lines 71-73, please rewrite to be more clear

Material and Methods:

Page 4 line 127: provide the source of cool-white fluorescent lamps

Page 5 line 160, what is the chemical composition of the buffer, and list the mixture of biogenic amines

Page 5 line183: some of the abbreviations here did not have a full definition before in the manuscript, please make sure that any abbreviation is associated with the full name ant the first mention in the manuscript.

Page 6 line 216: define FLD

Results and Discussion:

In this section change ‘orders of magnitude’ to fold for examples instead of saying two orders of magnitude it should be ‘two-fold’

Section 3.1.1: the authors are required to talk about their results first then mention the discussion to tell the reader what they found first, here I feel that I am reading a review article not a research article. The apply to section 3.1.2. and section 3.1.3.

Figures 2, 3, 4 and 5, should be merged into one figure with one four panels A, B, C, and D for easy comparison with the following caption “Figure x. Content of the polyamines agmatine (AGM), putrescine (PUT), cadaverine (CAD), spermidine (SPD) and spermine (SPM) in seeds, sprouts and microgreens of a) lentil, b) fenugreek, c) alfalfa and d) daikon radish. Data is presented on a logarithmic scale and expressed on a dry weight basis. The content of a given polyamine in seeds, sprouts and microgreens differs significantly if it is marked with different letters. There no need to keep the values on the top of each column just keep the letters.

Page 13 line: Figures A1-A2, there is no such Figures A1-A2 in the manuscript, I assume these in the appendix, please correct the figure number in the appendix.

Appendix:

First adjust the figure number to match the text

Figure 1, 2 and 3 should be merged into one figure (Figure A1) with three panels A, B and C

Figure 4, 5 and 6 should be merged into one figure (Figure A2) with three panels A, B and C

Page 17 line 496: remove ‘and’

Author Response

Response to Reviewer 1 Comments

Comments and Suggestions for Authors

The authors carried out the current study entitled “Accumulation of agmatine, spermidine and spermine in sprouts and microgreens of alfalfa, fenugreek, lentil and daikon radish “ to 1) determine the polyamine content in seeds, sprouts and microgreens of three legumes and one cruciferous plant, 2) evaluate whether microgreens are nutritionally superior over sprouts in terms of polyamine content and 3) evaluate the enzymatic potential of sprouts to degrade undesired biogenic amines. The study is on a topic of relevance and general interest to the readers of the journal. I found the paper to be overall well prepared and felt confident that the authors performed careful and thorough experiment and spectral processing. I have several significant concerns about the presentation of the data that should be addressed.

 Abstract:

Point 1: Authors need to avoid using personal pronoun (eg. We, our ..etc), please apply this over a;; the manuscript.

Response 1: In revised manuscript passive voice is used (once in the abstract and additional two times in the main text). Sentences with “our” were rewritten and “our” was replaced by “current”.

Point 2: Line 16-18: change of PUT to PUT, of SPD to SPD, and of SPM to SPM

Response 2: The prepositions “of” was omitted as suggested.

 Keywords:

Point 3: I recommend the author to use keywords that are differs from the title words

Response 3: Some keywords (Sprouts; Microgreens; Agmatine; Spermidine; Spermine) were omitted and new keywords were introduced (Biogenic amines; Germination; Medicago sativa; Trigonella foenum-graecum; Lens culinaris; Raphanus sativus).

Introduction:

Point 4: Table 1: some statements need to be supported by reference.

Response 4: Statements in the Table 1 are supported by 24 references. They are not included in the footnote but are instead listed as a regular reference in the section “References”. Within the Table 1 only the statements related to the biosynthesis of polyamines, which are generally well described in many literature sources, are not supported by explicit references.

Point 5: Some sentences are very long page 2 line 63-67 or confusing page 2 lines 71-73, please rewrite to be more clear.

Response 5: The long sentence (page 2 lines 63-67 in the original manuscript) was transformed into three shorter sentences. The sentence (page 2 lines 71-73 in the original manuscript) was rewritten.

 Material and Methods:

Point 6: Page 4 line 127: provide the source of cool-white fluorescent lamps

Response 6: The source and lamp details are included in the revised version of the manuscript.

Point 7: Page 5 line 160, what is the chemical composition of the buffer, and list the mixture of biogenic amines

Response 7: Two sentences (lines 168 -170 in the original manuscript) where composition of buffers was already described are introduced earlier in the text of the revised manuscript. Composition of the mixture of biogenic amines was also added.

Point 8: Page 5 line183: some of the abbreviations here did not have a full definition before in the manuscript, please make sure that any abbreviation is associated with the full name ant the first mention in the manuscript.

Response 8: Abbreviations for tyramine (TYR), tryptamine (TRP) and histamine (HIS) were introduced at their first mentions in the manuscript.

Point 9: Page 6 line 216: define FLD

Response 9: Abbreviation for fluorescence detector (FLD) was omitted.

Results and Discussion:

Point 10: In this section change ‘orders of magnitude’ to fold for examples instead of saying two orders of magnitude it should be ‘two-fold’

Response 10: Orders of magnitude are generally used to make very approximate comparisons. If two numbers differ by one order of magnitude, one is about ten times larger than the other. If they differ by two orders of magnitude, they differ by a factor of about 100.

Since there were very large difference in the content of polyamines in sprouts, seeds and microgreens the notation “order of magnitude” is in our opinion justified. Something that is for two orders of magnitude different could be in “-fold” notation ninety five-fold or one hundred and eleven-fold.

Point 11: Section 3.1.1: the authors are required to talk about their results first then mention the discussion to tell the reader what they found first, here I feel that I am reading a review article not a research article. The apply to section 3.1.2. and section 3.1.3.

Response 11: We agree with the reviewer that there was lack of actual experimental data in the text. In the revised manuscript the experimental data in the sections 3.1.1., 3.1.2., 3.1.3. and 3.1.4. are introduced before discussion of the results and comparison with the work of other authors.

Point 12: Figures 2, 3, 4 and 5, should be merged into one figure with one four panels A, B, C, and D for easy comparison with the following caption “Figure x. Content of the polyamines agmatine (AGM), putrescine (PUT), cadaverine (CAD), spermidine (SPD) and spermine (SPM) in seeds, sprouts and microgreens of a) lentil, b) fenugreek, c) alfalfa and d) daikon radish. Data is presented on a logarithmic scale and expressed on a dry weight basis. The content of a given polyamine in seeds, sprouts and microgreens differs significantly if it is marked with different letters. There no need to keep the values on the top of each column just keep the letters.

Response 12: Figures 2-5 were joined into Figure 2a-d as suggested. Values on the top of each column were introduced due to the logarithmic scale in order to point out to the very large differences in the content of particular polyamine in seeds, sprouts and microgreens. By omitting the numbers, the readers could in our opinion »underestimate« those differences.

Point 13: Page 13 line: Figures A1-A2, there is no such Figures A1-A2 in the manuscript, I assume these in the appendix, please correct the figure number in the appendix.

Response 13: The numbers of figures were corrected as suggested by reviewer and in accordance with the instructions for authors.

 Appendix:

Point 14: First adjust the figure number to match the text

Response 14: Figures from appendix are referred properly in the main text of the revised manuscript.

 Point 15: Figure 1, 2 and 3 should be merged into one figure (Figure A1) with three panels A, B and C

Response 15: Figures 1, 2 and 3 were joined to Figure A1a-c.

Point 16: Figure 4, 5 and 6 should be merged into one figure (Figure A2) with three panels A, B and C

Response 16: Figures 4, 5 and 6 were joined to Figure A2a-c.

Point 17: Page 17 line 496: remove ‘and’

Response 17: “and” was omitted.

Reviewer 2 Report

The manuscript foods-770252 "Accumulation of agmatine, spermidine, and spermine in sprouts and microgreens of alfalfa, fenugreek, lentil and daikon radish" submitted to the MDPI Foods Editorial Office concerns a very important subject from the point of view of plant material and applied methods. The problem of the production of sprouted seeds (sprouts) and seedlings (microgrens) in the aspect of their value for human health undertaken by the author is extremely important, and the results obtained by the research team are really valuable. The authors undertook a comparison of the polyamine profile during germination (4 days) and further developmental phases (another 10 days) using Medicago sativa, Trigonella foenum-graecum, Lens culinaris, Raphanus sativus for either sprouts or microgreens experimental production. Additionally, their enzymatic potential was checked in order to verify the ability to degage some undesired levels of biogenic amines.

It is worth emphasizing the work undertaken to outline the research problem raised in the manuscript in the introduction of the work (a synthetic summary of relevant information in Table 1 deserves to be highlighted here), clear specification of the objectives of undertaken experiments, then the results obtained and their interpretation. The methodology used is appropriate. A well-selected references with well used citations in the text.

All the more surprising the unexpected shortcoming in the definition of hypocotyl and epicotyl (lines 251-254). Epicotyl does not develop into leaves - leaves develop from apical bud. Similarly, hypocotyl being part of a stem below the cotyledons and directly above the root, does not develop into a root!

I am also wondering if it is worth trying to conduct a comprehensive analysis of the results obtained. Perhaps it may be worth rethinking it while analyzing your future results?

Author Response

Response to Reviewer 2 Comments

Comments and Suggestions for Authors

The manuscript foods-770252 "Accumulation of agmatine, spermidine, and spermine in sprouts and microgreens of alfalfa, fenugreek, lentil and daikon radish" submitted to the MDPI Foods Editorial Office concerns a very important subject from the point of view of plant material and applied methods. The problem of the production of sprouted seeds (sprouts) and seedlings (microgreens) in the aspect of their value for human health undertaken by the author is extremely important, and the results obtained by the research team are really valuable. The authors undertook a comparison of the polyamine profile during germination (4 days) and further developmental phases (another 10 days) using Medicago sativa, Trigonella foenum-graecum, Lens culinaris, Raphanus sativus for either sprouts or microgreens experimental production. Additionally, their enzymatic potential was checked in order to verify the ability to degarade some undesired levels of biogenic amines.

It is worth emphasizing the work undertaken to outline the research problem raised in the manuscript in the introduction of the work (a synthetic summary of relevant information in Table 1 deserves to be highlighted here), clear specification of the objectives of undertaken experiments, then the results obtained and their interpretation. The methodology used is appropriate. A well-selected references with well used citations in the text.

Point 1: All the more surprising the unexpected shortcoming in the definition of hypocotyl and epicotyl (lines 251-254). Epicotyl does not develop into leaves - leaves develop from apical bud. Similarly, hypocotyl being part of a stem below the cotyledons and directly above the root, does not develop into a root!

Response 1: We apologize for incorrect description of hypocotyl and epicotyl. In the revised manuscript, epicotyl and hypocotyl were simply described as the region above and below cotyledons: “A separate analysis of the polyamines in the epicotyl (part of seedling above cotyledons) and the hypocotyl (part of seedling below cotyledons) of sprouts showed that the polyamine content in the epicotyl was by an order of magnitude lower (results not shown).”

Point 2: I am also wondering if it is worth trying to conduct a comprehensive analysis of the results obtained. Perhaps it may be worth rethinking it while analyzing your future results?

Response 2: We agree that this is an interesting topic and therefore we are continuing our work on the transformation of polyamines during germination. Combination of exogenously applied polyamines into the growth media and salt stress seems to lead to the higher polyamine content in microgreens of grasses (wheat, barley), where growth in the Hoagland solution did not results in such increase as in legumes and radish.

Round 2

Reviewer 1 Report

No further comments.